

# Seasonal streamflow forecasts in the Ahlergaarde catchment, Denmark: effect of preprocessing and postprocessing on skill and statistical consistency

Diana Lucatero[1], Henrik Madsen[2], Jens C. Refsgaard[3], Jacob Kidmose[3], Karsten H. Jensen[1]

[1]Department of Geosciences and Natural Resource Management, University of Copenhagen, Copenhagen, Denmark

[2] DHI, Hørsholm, Denmark

[3] Geological Survey of Denmark and Greenland (GEUS), Copenhagen, Denmark

*Correspondence to:* Diana Lucatero (diana.lucatero@ign.ku.dk)

**Abstract.** In the present study we analyze the effect of bias adjustments in both meteorological and streamflow forecasts on skill and reliability of monthly average streamflow and low flow forecasts. Both raw and pre-processed meteorological seasonal forecast from the European Center for Medium-Range Weather Forecasts (ECMWF) are used as inputs to a spatially distributed, coupled surface – subsurface hydrological model based on the MIKE SHE code in order to generate streamflow predictions up to seven months in advance. In addition to this, we postprocess streamflow predictions using an empirical quantile mapping that adjusts the predictive distribution in order to match the observed one. Bias, skill and statistical consistency are the qualities evaluated throughout the forecast generating strategies and we analyze where the different strategies fall short to improve them. ECMWF System 4-based streamflow forecasts tend to show a lower accuracy level than those generated with an ensemble of historical observations, a method commonly known as Ensemble Streamflow Prediction (ESP). This is particularly true at longer lead times, for the dry season and for streamflow stations that exhibit low hydrological model errors. Biases in the mean are better removed by postprocessing that in turn is reflected in the higher level of statistical consistency. However, in general, the reduction of these biases is not enough to ensure a higher level of accuracy than the ESP forecasts. This is true for both monthly mean and minimum yearly streamflow forecasts. We highlight the importance of including a better estimation of the initial state of the catchment, which will increase the capability of the system to forecast streamflow at longer leads.

## 1 Introduction

Seasonal streamflow forecasting encompasses a variety of methods that range from purely data based to entirely model based or hybrid methods that exploit the benefits of each (Mendoza et al., 2017). Data driven methods find empirical relationships between streamflow and a variety of predictors and use these to derive forecasts for the upcoming seasons. Different predictors can be used depending on the relative importance they have on the regional hydroclimatic conditions. Predictors that have been used include large scale climate indicators such as el Ninio, (Schepen et al., 2016; Shamir, 2017; Wang et al., 2009), precipitation and land temperature (Córdoba-Machado et al., 2016), state of the catchment in the form of streamflow, soil moisture, groundwater storages or snow storages that can be derived either by the use of a hydrological model, therefore the term 'hybrid' (Robertson et al., 2013; Rosenberg et al., 2011), or by means of observed antecedent conditions (Robertson and Wang, 2012).

Model based systems include a hydrological model in the forecasting chain. Differences between forecasting frameworks may arise in the forcings, the initialization framework and/or the hydrological model structure and parameters. Focusing on the forcing, one can either use observed meteorology from previous years, a method that is commonly known as Ensemble Streamflow Prediction (ESP) (Day, 1985), or outputs from, use General Circulation Models (GCM) outputs (Crochemore et





al., 2016; Wood et al., 2002, 2005; Wood and Lettenmaier, 2006; Yuan et al., 2011, 2013, 2015, 2016). In principle, the latter should be more suitable in providing skilful forecasts as GCMs are able to capture the evolving chaotic behavior of the atmosphere, whereas the ESP approach assumes that what has been observed in the past can be used as a proxy for what will happen in the future, an assumption that requires stationary climate conditions. On the other hand, the perceived lack of

reliability GCMs have in forecasting atmospheric patterns at long lead times preclude them for use in weather impacted sectors (Bruno Soares and Dessai, 2016).

This is the reason why pre- and postprocessing may be performed when using GCM forecasts to force a hydrological model in order to eliminate biases intrinsic in climate and hydrological models. In the context of this study, preprocessing refers to any method that improves the forcings, i.e. precipitation and temperature, used in the hydrological forecasting system.

Postprocessing refers to the improvements done to the outputs of the hydrological model, e.g. streamflow. In this respect, postprocessing also corrects errors in hydrological models that cannot be eliminated through calibration (Shi et al., 2008; Yuan et al., 2015; Yuan and Wood, 2012).

A couple of studies have quantified the effects on streamflow skill by preprocessing either seasonal (Crochemore et al., 2016) or medium range forecasts (Verkade et al., 2013) forecasts. Other studies have assessed the efficiency of

postprocessing streamflow forecasts only (Bogner et al., 2016; Madadgar et al., 2014; Ye et al., 2015; Zhao et al., 2011;Wood and Schaake, 2008). To our knowledge, only Roulin and Vannitsem, (2015); Yuan and Wood, (2012) and Zalachori et al., (2012) have compared the additional gain in skill of doing both pre- and postprocessing. The previous studies have shown that improvements made by preprocessing the forcings do not necessarily translate into improvements in streamflow forecasts (Verkade et al., 2013; Zalachori et al., 2012). Improvements are larger when postprocessing is done,

and a combination of pre- and postprocessing provides the best results (Yuan and Wood, 2012; Zalachori et al., 2012). To our knowledge, only Yuan and Wood, (2012) have made this evaluation in the context of seasonal forecasting.

The present study focuses on the following aspects: (i) the evaluation of the use of a GCM to generate seasonal forecasts, (ii) the study of the effect that pre- and postprocessing have on streamflow forecasts 1-7 months ahead, and (iii) the effect of hydrological model biases in forecast skill evaluations. This is done by a combination of the following methodological

choices. First, we make use of seasonal meteorological forecasts from ECMWF System 4 (Molteni, et al., 2011). Secondly, the hydrological simulations are based on an integrated physically-based and spatially distributed model based on the MIKE SHE code. Thirdly, our evaluation focuses on three forecast qualities: bias, skill and reliability, with skill measured using ESP as a reference and focusing on both accuracy and sharpness. Finally, the focus here is to evaluate forecasts of monthly average streamflow throughout the year and low flows during the summer of a groundwater dominated catchment located in

a region where seasonal forecasting is a challenging endeavor. The following questions are then addressed:

(1) How does GCM generated forecasts compare to that of the ESP approach?

(2) What is the effect of pre- and postprocessing on streamflow forecasts in terms of bias, skill and statistical consistency. And more specifically, is there one single approach, or a combination of several that reduces the bias and augments skill and statistical consistency?

(3) What is the effect that hydrological model bias has on the evaluation of pre- and postprocessed streamflow forecasts?

**2 Data and Methods**

The following sections give a description of the methodology followed in this study. A graphical depiction of the steps carried out can be seen in Fig. 1.





### 2.1 Area of study, observational data and hydrological model

The present study is carried out for the Ahlergaarde catchment located in West-Jutland, Denmark (Fig. 2). The catchment covers an area of 1044 km$^2$ and is located in one of the most irrigated zones in Denmark with 55% of the area covered with agricultural crops such as barley, grass, wheat, maize and potatoes. The remaining area is distributed as follows: grass

(30%), forest (7%), heath (5%), urban (2%), and other (1%) (Jensen and Illangasekare, 2011).

The climatology of the area is shown in Fig. 3. Climate in the Ahlegaarde region is mainly driven by its proximity to the sea towards the west. The mean annual precipitation for the period 1990-2013 is 983 mm. The hydrology of the catchment is groundwater dominated due to the high permeability of the top geological layer, which consists mainly of sand and gravel. The 1990-2013 average monthly streamflow has a maximum in January (60 mm) and a minimum during the summer (25

mm). Another consequence of the geological composition of the surface layer is that overland flow rarely happens. A more detailed description of the geology of the area can be found in Kidmose et al., (*unpublished work*).

Precipitation (P), temperature (T) and reference evapotranspiration ($E_{T0}$) data are retrieved from the Danish Meteorological Institute (DMI), which covers Denmark with a 10 km grid resolution for P and a 20 km resolution for T and $E_{T0}$ (Scharling and Kern-Hansen, 2012) with P corrected for systematic under catch due to wind effects (Stisen et al., 2011, 2012).

Streamflow observations are taken from the Danish Hydrological Observatory (HOBE) (Jensen and Illangasekare, 2011).

The hydrological simulations for this study are based on a physically-based, spatially distributed, coupled surface-subsurface model that simulates the main hydrological processes such as evapotranspiration, overland flow, unsaturated, saturated and stream flows and their interactions. The model is based on the MIKE SHE code (Graham and Butts, 2005). Groundwater flow is described by the governing equation for three-dimensional groundwater flow based on Darcy's law. Drain flow is

considered when the groundwater table exceeds a drain level. Surface water flow in streams is simulated by a one-dimensional channel flow model based on kinematic routing, while a two dimensional diffusive wave approximation of the St. Venant equations is used for overland flow routing. Finally, a two layer approach is used for the simulations of unsaturated flow and evapotranspiration (Graham and Butts, 2005). The horizontal numerical discretization is 200 meters, whereas the vertical discretization is based on six numerical layers whose dimension depends on the geological stratigraphy.

Model parameters were calibrated against groundwater head and discharge using an automated optimizer, PEST (Parameter Estimation) version 11.8 (Doherty, 2016) for the 2006-2009 period. Parameters to be calibrated were selected based on a sensitivity analysis study. These are: hydraulic conductivities for ten geological units, specific yield, specific storage, drain time constant, detention storage, river-groundwater conductance and root depth of ten vegetation types. The reader is referred to Zhang et al., (2016) and Kidmose et al., (*unpublished work*) for further details on the calibration procedure.

### 2.2 Forecast generation: GCM-based and ESP

As seen in Fig. 1, P, T and $E_{T0}$ forecasts are taken from the ECMWF System 4 (RAW), preprocessed ECMWF System 4 (LS and QM), and historical observations (ESP). The European Center for Medium-Range Weather Forecasts (ECMWF) offers a seasonal forecasting product that currently is in its version number 4 (Molteni et al., 2011). An attempt to reduce the biases intrinsic in ECMWF System 4 led to what we refer to as preprocessed forecasts. The reader is referred to Lucatero, (*in*

*preparation*) for details of the evaluation of both ECMWF System 4 and preprocessed forecasts for Denmark. The spatial resolution of the raw forecasts is 0.7 degrees in the latitude and longitude that were interpolated to a 10 km grid to match the resolution of the observed precipitation grid. For the Ahlergaarde catchment, a total of 24 grid points were extracted that cover the study area for the 1990-2013 period, leading to a sample size of 24 years. $E_{T0}$ is computed using the Makkink formulation (Hendriks, 2010) that takes as inputs T and incoming shortwave solar radiation from the ECMWF System 4

forecasts.





Raw and preprocessed forecasts are initialized on the first day of each calendar month and have a lead time of 7 months with a daily temporal resolution. The number of ensembles varies with month, 15 for January, March, April, June, July, September, October and December, and 51 for the remaining months. ESP forcings are taken from the observation record with each year acting as an ensemble member. The values are taken from the start of each calendar month, with a 7-month

lead time in order to match the lead time of the ECMWF System 4 forecasts. The year to be forecasted is withdrawn from the ensemble. Thus, the number of ensemble members for the ESP is 23. Both the ECMWF System 4 generated forecasts and ESP share the same hydrological initial conditions for forecasts initiated on the same month. These are computed from a spin-up run starting in January 1990 and up until the time of the forecast initialization. Forecasts are then run on a daily basis up to seven months.

**2.3 Preprocessor and Postprocessor**

Preprocessed forcings for the hydrological model were retrieved from Lucatero, et al., (*in preparation*). The authors used two well-known bias correction techniques, namely, Linear Scaling/Delta Change (hereafter, LS) and Quantile Mapping (QM). In LS the ensemble is adjusted with a scaling factor, either by multiplication (for P and $E_{T0}$) or addition (T). The scaling factor is computed as the ratio or difference between the averages of the ensemble mean and the observed mean for a

specific month, lead time and location, with the sole purpose of adjusting the mean.

QM matches the quantiles of the ensemble distribution with the quantiles of the observed distribution in the following way:

$$f_{k,i}^{*} = G_i^{-1}\left(F_i\left(f_{k,i}\right)\right) \tag{1}$$

where $G$ and $F$ represent the observed and the ensemble distribution functions, respectively, for forecast-observation pair $i$, for $i = 1, \cdots, M$ with $M$ being the number of forecast-observation pairs. $f_{k,i}$ represents ensemble member $k$, $k = 1, \ldots, N$

where $N$ is the ensemble size, and $f_{k,i}^{*}$ represents the corrected ensemble member $k$. $F$ is an empirical distribution function trained with all ensemble members at a given month for a given lead time and location on a leave-one-out-cross-validation mode. For example, for a forecast of target month April initialized in February, $F$ is computed using all ensemble members, comprising 30 (days) times 23 (number of years in the training sample minus the year to be corrected) times the ensemble size of that particular month (15 or 51). The same is done for $G$. Linear extrapolation is applied to approximate the values

between the bins of $F$ and $G$ and to map ensemble values and quantiles that are outside the training sample.

QM is the only method used for postprocessing in the present study as no striking differences in both bias and skill were found between LS and QM in Lucatero, et al., (*in preparation*). Moreover, QM shows more satisfactory results for the correction of forecasts in the lower tail of the distribution and for correcting forecasts that also exhibit underdispersivity Lucatero, et al., (*in preparation*).

**2.4 Performance metrics**

To evaluate first the raw forecasts and the improvement after preprocessing and postprocessing, we check for four qualities: bias, skill in regards to accuracy and sharpness, and statistical consistency. Bias is the measure of under or overestimation of the mean of the ensemble in comparison with the observed values (Yapo et al., 1996):



$$PBias = \left( \frac{\sum\limits_{i=1}^{M} \bar{f}_i}{\sum\limits_{i=1}^{M} y_i} - 1 \right) \cdot 100 \qquad (2)$$

where $\bar{f}_i$ and $y_i$ represent, respectively, the ensemble mean and the observed values for forecast-observation pair $i$ of a particular month, lead time and location. If the value in Eq. (2) is negative, we have an underprediction, and conversely an overprediction if the value is positive.

Secondly, we compute the continuous rank probability score CRPS (Hersbach, 2000) as a general measure of the accuracy of the forecasts. The computation of the score is as follows

$$CRPS = \frac{1}{M} \sum_{i=1}^{M} \int_{-\infty}^{\infty} \left[ P_i(x) - H(x - y_i) \right]^2 dx, \qquad (3)$$

where $P_i(x)$ represents the CDF of the ensemble for forecast-observation pair $i$, $H(x - x_i)$ is the Heaviside function that takes the value 0 when $x < y_i$ or 1 otherwise. $y_i$ is the verifying observation of forecast-observation pair $i$ of $M$ forecast-

observation pairs. Sharpness for forecast-observation pair $i$ is measured as the difference between the 25% and the 75% percentiles. The average of these differences along the forecast-observation record is then used as a measure of sharpness. Both the CRPS and the sharpness score are then given in the units of the variable of interest, i.e. m³/s for streamflow. Moreover, both scores of a system that is perfect is zero. A skill score can be then computed in the following manner

$$Skill = 1 - \frac{Score_{sys}}{Score_{ref}} \qquad (4)$$

where, for the present study, $Score_{sys}$ is the score of streamflow forecasts generated either with raw, preprocessed ECMWF System 4 or the postprocessed forecasts, $Score_{ref}$ is the score value of our reference system, the ESP. The range of the skill score in Eq. (4) is from $-\infty$ to 1, and values closer to 1 are preferred. Negative values indicate that, on average, our system does not manage to beat the ESP. Hereafter, we denote the skill with respect to accuracy as CRPSS and the skill in terms of sharpness as SS. In order to evaluate the statistical significance of the differences of skill between GCM generated forecasts

and ESP, we use a two-sided Wilcoxon-Mann-Whitney test (WMW-test) at the 5% significance level (see Hollander et. al., 2014).

Finally, in order to evaluate the statistical consistency between predictive and observed distribution functions we use the Probability Integral Transform (PIT) diagram. The PIT diagram is the cumulative distribution function (CDF) of $z_i = P(X \leq y_i)$, where $z_i$ is the value of the cumulative distribution function that the observed value attains within the

ensemble distribution for each forecast-observation pair $i$. For a forecasting system to be statistically consistent, meaning that the observations can be seen as a draw of the predictive CDF, the CDF of the $z_i's$ should be close to the CDF of a uniform distribution in the [0,1] range. Deviations from the uniform distribution signifies bias in the ensemble mean and spread as explained in Appendix 1. In order to make the test for uniformity formal, we make use of the Kolmogorov confidence bands. The bands are two straight lines, parallel to the 1:1 diagonal and at a distance $q(\alpha)/\sqrt{N}$ where $q(\alpha)$ is a coefficient that

depends on the significance level of the test, i.e., $q(\alpha = 0.05) = 1.358$ (see Laio and Tamea, 2007; D'Agostino and Stephens,





1986) and $N$ is again the number of forecast-observation pairs. The test for uniformity is passed if the CDF of the $z_i's$ lies within these bands.

### 2.5 Low flow forecasting

Low flow forecasts can be used for optimizing groundwater extractions for irrigation. The years for which the predicted low flows are above the prescribed minimum can be exploited and utilized for crops with a higher irrigation demand that may increase economic returns. Here we focus on forecasts initiated in April. For the purposes of this study, low flows are defined as the flow of the day with the minimum yearly discharge ($m^3$/s) that usually happens during July to September (Fig. 3). Low flow forecasts are evaluated using the same skill scores as for monthly flow forecasts. Studies that have focused their attention to situations of low flow or hydrological drought in the context of seasonal forecasting exist: Demirel et al., (2015); Trambauer et al., (2015).

### 3 Results

#### 3.1 Hydrological model evaluation

Figure 4 shows the results for simulated streamflow at the upstream station 21 and the downstream station 82 for daily values during the period from 2000-2003. As a preliminary evaluation, we computed the percent bias (PBias) and the Nash-Sutcliffe model efficiency coefficient (NSE) for the complete observed-simulated record (1990-2013). There is, in general, a good agreement in timing between observed and simulated values. The visual inspection of the hydrographs reveal, however, an amplitude error that is more pronounced at the upstream station 21. Evidence for this is also reflected by the high values of bias and the negative NSE for this station (NSE = -0.85). Furthermore, a scatter plot of simulated and observed low flows for the 24 years shows an overestimation of the low flows that is more pronounced at the upstream station (Fig. 4). At the outlet station 82 there is a better behavior in terms of bias and NSE, with an overestimation of only 1.7% and a NSE of 0.73. Moreover, for this station there is a better agreement in both the high and low flows through the year. The latter can be verified by looking at the scatter plot of the low flows (Fig. 4), with the majority of points lying close to the 1:1 diagonal.

Due to the poor performance in the upstream station 21, in the following sections (3.2 - 3.4) we will discuss the skill and consistency of the different approaches for forecast improvement with a focus on the outlet station only. The large biases in the upstream station, combined with the structural biases of the meteorological forecasts seem to inflate the skill of the streamflow forecasts. This will be discussed in Section 3.5.

#### 3.2 Streamflow forecasts forced with raw meteorological forecasts

The bias and skill of the monthly streamflow forecasts forced with raw ECMWF forecasts are shown in the first row of Fig. 5. The X-axis represents the different lead times in months, while the Y-axis represents the target month. For example, the bias of November with lead time 2 represents the value of bias for a forecast in that month initiated on October 1. This bias is in the [-30,-20%] range. In general, the absolute bias increases with lead time, and usually moves from an overprediction (or mild underprediction) to a large negative bias at longer lead times.

Figure 5 also shows the skill of accuracy and sharpness. The months with statistically significant differences in skill between the ESP and ECMWF System 4 forecasts are represented with a black circle. There is a connection between bias and skill of accuracy in the sense that months with a higher bias tend to be the ones with lower or non-existent skill. The opposite also holds, months with milder bias tend to be the months where the forecast is improving over the reference forecast to a higher degree. This is by no means surprising, as the CRPS penalizes forecasts that have biases.



The CRPSS is negative, except for some months during winter and at short lead times for which a forecast generated with raw ECMWF System 4 forcings improves accuracy up to 40% compared to ESP. As for the case of bias, skill depends on lead time, reaching its most negative values for forecasts generated 7 months in advance. One important feature is the high skill that a forecast generated with ECMWF raw forcings has in terms of sharpness. Figure 5 shows that this quality is present in the majority of target months and lead times. Note, however, that sharpness is only a desirable property when biases are low. In this case, the width of the raw forecasts is smaller than that of the ESP, indicating overconfidence, when biases are high.

Statistical consistency of the raw forecasts is visualized on the first column of the PIT diagrams in Fig. 6 for winter and summer (first and second row, respectively) at lead time 1. Kolmogorov confidence bands are also plotted for a graphical test of uniformity at the $\alpha = 0.05$ level. For the sake of brevity, the remaining seasons and lead times are not shown. For the particular seasons and lead time shown, statistical consistency seems to be achieved only for the wettest months (Dec-Feb). The explanation for this particular behavior will be given in Sect. 3.5. Early spring and November forecasts are also able to pass the uniformity test (not shown). Summer forecasts together with late spring and autumn months (May, September and October) show a significant underprediction, which prevent them to pass the uniformity test. Statistical consistency appears to get worse as lead time increases, in accordance with the deterioration of the bias in Fig. 5.

### 3.3 Streamflow forecasts forced with preprocessed meteorological forecasts

The second and third rows of Fig. 5 show the bias and skill of streamflow forecasts generated with preprocessed forcings from ECMWF System 4 using the LS and the QM method, respectively.

Several conclusions can be drawn when comparing forecasts using the preprocessed and raw forcings. First, the biases are clearly improved, especially for longer lead times. For example, for October forecasts from lead time 3 to 7 months, biases are reduced from the [-40,-30%] to the [-20,10%] range for LS and to the [-15,20%] range for QM. There are, however, no obvious differences between the two preprocessing methods, which seem to perform equally well in reducing biases. Secondly, three features on accuracy are seen. The first one is that, also for accuracy, there are no obvious differences in skill between the two preprocessing methods. Furthermore, there seems to be a reduction of skill for the winter months and March at the first month lead time. These months are the only ones with a statistically significant skill using the raw forecasts. This feature is a consequence of the reduction of the forcing biases, situation that will be further discussed in Sect. 3.5. The last feature is that the improvement of the forcings can help reducing the negative skill in streamflow forecasts. For example, April to November forecasts at longer lead times, generated with raw ECMWF System 4 forcings, exhibit a highly negative and statistically significant skill, sometimes lower than -1.0. Streamflow forecasts generated with preprocessed forcings for those months tend to have a neutral skill implying that their accuracy is not different from the accuracy obtained with ESP. The final conclusion is related to sharpness. As we can see in Fig. 5, streamflow forecasts generated with preprocessed forcings have an ensemble range that is wider than the reference forecasts.

The second and third columns in Fig. 6 show the PIT diagrams of streamflow forecasts generated with preprocessed forcings for the winter and summer forecasts at the first month lead time. The statistical consistency for the winter months seems to be worsening, in comparison to the consistency of the forecasts generated with raw forcings. The same degree of deterioration is seen for both preprocessing methods. This is caused by compensational errors that will be further discussed in Sect. 3.5. Besides from that particular season, improvements in consistency after preprocessing can be seen during the autumn (not shown), and August, although to a lesser degree. For spring and early summer forecasts, the same level of consistency is observed for both the raw and preprocessed forecasts. At longer lead times, the benefit of preprocessing for statistical consistency is clearer, most of the months pass the uniformity test.




### 3.4 Postprocessed streamflow forecasts

The final step in the analysis is the postprocessing of streamflow forecasts generated with raw and preprocessed ECMWF System 4 forcings. Fig. 7 shows the verification results that can be directly compared to the results in Fig. 5.

The first column in Fig. 7 shows a clear reduction of the absolute bias compared to the raw and preprocessed generated forecasts. Bias lies within the range [-10,10%], for all months and lead times. Furthermore, the majority of the CRPSS for all months and lead times are positive, while a small negative skill is seen during the autumn. Note, however, that the differences in accuracy between ESP and the postprocessed forecasts are only significant at the 5% level for few target months and lead times. In general, there seems to be a worsening of the sharpness after postprocessing (Fig. 5). However, this deterioration is lower when comparing preprocessed versus postprocessed forecasts. Furthermore, the degree of the deterioration varies according to the target month. For example, summer months (June and July) exhibit a larger deterioration of sharpness, i.e., forecast spread is larger than that of the ESP. On the other hand, forecasts for late autumn and early December appear to be narrower than ESP forecasts after postprocessing.

Figure 8 shows the PIT diagrams for the summer and winter seasons at the first month lead time of the postprocessed streamflow forecasts. The plot can be directly compared to Fig. 6. As seen from the PIT diagram, all months in those seasons pass the uniformity test, indicating that after postprocessing, the observations can be considered as random samples of the predictive distribution. The remaining PIT diagrams for spring and autumn and lead times 2-7 months (not shown in Figure 8) show that statistical consistency is present for all months and lead times. At longer lead times, the CDFs of the $z_i's$ are closer to the 1:1 diagonal. This is achieved due to two factors: (i) the additional reduction of bias after postprocessing, and (ii) the worsening of sharpness for long lead times where the larger ensemble spread encloses a larger portion of observed values.

### 3.5 Effect of hydrological model bias in skill evaluations

As mentioned in Sect. 3.1., hydrological model biases, which are larger for the upstream station 21 (Fig. 4), combined with the structural biases in GCMs, can lead to a situation with a high skill resulting from compensational errors providing "the right forecast for the wrong reasons". In order to illustrate this point, Fig. 9a-9b show the CRPSS for, respectively, station 21 with large bias (PBias = 48%, Fig. 4) and station 82 with small bias (PBias = 1.7%, Fig. 4). The figure shows CRPSS for forecasts generated with raw ECMWF forcings and preprocessed forcings with the LS method for the target months Jan-Dec at lead time 4 (e.g. January forecasts initiated in October). In addition to the comparison against observations, we also include a comparison against simulated streamflows (continuous run of the Ahlergaarde model with observed meteorological forcings, Fig. 4). This is done in order to remove the effect of hydrological model bias and hence focus the analyses on the biases coming from forcings alone.

The high skill against observed streamflows is more visible during the wettest months (November-April) for station 21 where hydrological model biases are highest (Fig. 4). Once the comparison is made against simulated streamflows, the high positive skill becomes highly negative (Fig. 9a). The deterioration of skill when compared against simulated streamflows is also seen at station 82 for Dec-March, although to a lesser extent (Fig. 9b). To illustrate why this happens, Fig. 9c and 9d show the monthly streamflow forecasts for all 24 years for target month December of forecasts initialized in September ( lead time 4). Both ESP and raw (Fig. 9c) and preprocessed (Fig. 9d) forecasts are shown, along with their respective skill scores of accuracy, when the comparison is made against observed (CRPSS) and simulated (CRPSS.s) values.

Figure 9c shows two issues. First, the large hydrological model bias that cause ESP to have a deviation from the observations, leading to a high CRPS for the reference forecast in Eq. (4). Secondly, for the winter months, precipitation



from the raw ECMWF System 4 forecasts exhibit a negative bias of around -15% (Lucatero, et. al., *in preparation*). This compensates the biased streamflow forecasts and results in a low $CRPS_{Sys}$ value in Eq. (4). The CRPSS then becomes positive and high (0.54). However, when the comparison is done against simulated values, the skill score becomes highly negative (CRPSS.s = -0.41). Once the biases in the forcings are removed (Fig. 9d), then the hydrological model bias takes

over, leading these forecasts to the same level as the ESP, increasing its CRPS, which in turn reduces the skill score (CRPSS = -0.04).

Stations like 21 could benefit the most from postprocessing that will remove hydrological model biases that calibration alone could not remove. This is illustrated with the visualization of the CRPSS of the different forecasts in Fig. 10a-10d. The comparison is made against observations. Figure 10b shows a reduction of the skill after the raw forcings have been

preprocessed, as a result of the compensation errors discussed above. However, once the hydrological biases are removed with postprocessing (Fig. 10c and 10d), skill is positive and significant throughout November to April. Note, however, that the high skill at this particular station is mainly driven by the poor performance of the reference ESP, due to the large bias of the hydrological model (Fig. 4). It is also worth noting the lack of differences in skill between Fig. 10c and Fig. 10d, showing that, for this particular location, a combination of preprocessing plus postprocessing is just as good as

postprocessing of the forecasts generated with raw forcings alone.

### 3.6 Low flow forecasting

In addition to the evaluation of the monthly streamflow forecasts, we have assessed whether the use of GCM forecasts can add value to the forecasting of annual low flows compared to the ESP. Figure 11 shows the low flow forecast, issued in April, at the outlet stations 82 for both the raw forecasts and the different pre- and postprocessing strategies. Black boxplots

represent the forecast generated using the raw outputs of the ECMWF System 4 (Fig. 11a), the preprocessed forecasts (Fig. 11c and 11e) and the postprocessed forecasts (Fig. 11b, 11d, 11f). The box-plots in the background (blue) represent the ESP forecasts and the red dots represent the yearly observed minimum discharges. When we look at Fig. 10a, several features can be highlighted. First, despite the underprediction of the raw generated forecasts and, to a lesser extent, the ESP forecasts of the highest minimum discharges in the 00s, the year-to-year variability is replicated well, i.e., low observed low flow values

have low, although biased, forecasted values, and high observed low flow values have, in general, high forecasted values. Secondly, even though the raw generated forecasts are sharper than the ESP by about 10% (SS = 0.11), they do not manage to beat ESP in terms of skill of accuracy (CRPSS = -0.14), i.e. they are overconfident.

Preprocessing meteorological forecasts seem to have a positive effect in low flow forecasting, reducing the CRPSS from -0.14 to -0.01 when using the LS preprocessor. This happens because of the loss of sharpness (from 0.11 to -0.11) that allow

the forecasts to better capture the higher low flows during the 00's. However, it is still difficult to beat the ESP. Postprocessing seems to have a similar effect, loss of sharpness, decrease in bias that allow the forecasts to capture the high low flows in the 00s and 10s. This situation, however, leads to a loss in skill in forecasting low flows in the 90s, leveling out the skill to a similar score (CRPSS = -0.12) as the forecasts generated with raw ECMWF forcings (CRPSS = -0.14). It thus seems that an attempt to reduce meteorological and hydrological biases through processing the forcings and/or the

streamflow will result in only a modest increase in skill of low flow predictions on average. ESP remains a reference forecast system difficult to beat.

### 4 Summary and conclusions

Seasonal forecasts of streamflows initiated in each calendar month for the 1990-2013 period were generated for a groundwater dominated catchment located in a region where seasonal atmospheric forecasting is a challenge. We analyzed

the bias and statistical consistency of monthly streamflow forecasts forced with ECMWF System 4 seasonal forecasts along





all calendar months throughout the year. In addition to this, we evaluated their accuracy and sharpness relative to that of the forecasts generated with an ensemble of historical meteorological observations, the ESP. Due to the systematic errors of GCM seasonal forecasts and errors in the hydrological model that calibration alone cannot defuse, both pre and postprocessing using two popular and simple correction techniques were used to remove them: LS and QM. Finally, we also

estimate the skill that the different forecast generation approaches have on forecasting the minimum yearly discharge.

Monthly streamflow forecasts generated with raw forcings are affected by biases both in the hydrological model and the forcing. This bias grows as lead time increases and also reflected in the shape of the PIT diagrams that exhibit a persistent underestimation of the forecast mean for the majority of the calendar months and lead times. GCM-based forecasts are, however, sharper than ESP forecasts, which combined with the high biases lead to a poorer accuracy than ESP forecasts. Our

results are directly comparable to those of Crochemore et al., (2016) as they also deal with seasonal streamflow forecasting with ECMWF System 4 for a region located not too far from our study area. They also found GCM-based forecast to be sharper than those of the ESP. However, biases intrinsic in the GCM are also propagated into the streamflow forecasts, which lead them to have a poorer accuracy. Overall, the GCM-based forecasts did not performed better that those based on ESP. Other studies that have compared the skill of GCM-based streamflow forecasts versus those based on ESP have found

similar results (Wood et al., 2005; Yuan et al., 2013). In the present study we found that this is especially difficult during the dry months and for streamflow stations that exhibit low hydrological model error. Furthermore, caution must be taken when hydrological model errors are high, as it may lead to erroneous evaluations of skill when hydrological model biases are neutralized by opposite GCM errors, e.g., forecasts of monthly streamflow during the winter.

Preprocessing of the forcings alone helped to reduce streamflow biases and reduce the negative skill at longer lead times.

The reduction of the under- or overestimation, lead to forecasts with a higher statistical consistency, for most of the months and lead times considered. This rather mild enhancement was also found by Crochemore et al., (2016). Moreover, postprocessing alone does a better job in removing biases in the mean, that in turn help to ameliorate issues with the statistical consistency. Ye et al., (2015) and Zalachori et al., (2012) also report the above behaviour, whereas Yuan and Wood, (2012) found a better correction of statistical consistency after both pre- and postprocessing. The removal of biases of

both forcings and hydrological model did not ensure a higher level of accuracy than the ESP, as demonstrated by the non-significant differences of accuracy between GCM-based forecasts and the ESP. This is also true for the low flow forecasts as mentioned above.

One must keep in mind, however, that the results presented here may depend on the catchment characteristics as well as on the climatic conditions of the study area. Seasonal meteorological forecasting is still a difficult task, particularly on regions

further away from the tropics, which in turn translate into the streamflow forecasts. One obvious omission of the study presented is the exploitation of storages in the form of snow, soil moisture or/and groundwater and taking advantage of the hydrological memory that may increase skill at longer leads. This has been the routine for snow dominated catchments in western U. S. by means of ESP (Wood and Lettenmaier, 2006). However, a preliminary evaluation of the relationship between groundwater levels in winter and low flows during the summer in the same catchment studied here showed that

relatively high correlations exist in large parts of the catchment (Kidmose, et al., *unpublished work*). Predictability attribution studies exist that quantify the sensitivity of the skill of a forecasting system relative to different degrees of uncertainty, either in the forcing or the initial conditions. Wood et al., (2016) developed a framework to detect where to concentrate on improvements, e.g., either the initial conditions, usually by means of data assimilation (Zhang et al., 2016), or the seasonal climate forecasts. This might shed light on, and possibly reinforce the hypothesis that for groundwater

dominated catchments and forecasting of low flows, initial conditions will have a higher influence on forecast skill at longer lead times (Paiva et al., 2012).



## 5 Appendix A. Interpretation of the PIT Diagram

Fig. A1 serves as a basis for the interpretation of the PIT diagrams. The figure is modified from Laio and Tamea, (2007). Four situations can arise: (i) overprediction, or positive bias in the mean when the CDF of the $z_i$'s lies above the 1:1 diagonal; (ii) underprediction, or negative bias in the mean when the CDF of the $z_i$'s lies below the 1:1 diagonal; (iii) overdispersion, or

underconfident forecasts (large forecasts) when a greater proportion of the values of the CDF lie on the middle ranges bins of the distribution; and (iv) underdispersion, or negative bias in spread (overconfidence) when a greater proportion of the values of the CDF lie on the tails of the distribution.

## 6 Acknowledgements

This study was supported by the project 'HydroCast - Hydrological Forecasting and Data Assimilation', Contract #0603-

00466B (http://hydrocast.dhigroup.com/) funded by the Innovation Fund Denmark. Special thanks to Florian Pappenberger for providing the ECMWF System 4 reforecast and Andy Wood and Pablo Mendoza for hosting the first author at NCAR.

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

**8 Figure Captions**

**Figure 1:** Diagram of generation of forecasts and verification procedures. RAW refers to the uncorrected ECMWF System 4 forecasts, while LS and QM refer to forecasts (either meteorological or hydrological) that are corrected using the Linear Scaling/Delta Change or Quantile Mapping method, respectively, for precipitation (P), temperature (T) and reference evapotranspiration ($E_{T0}$).

**Figure 2:** Location and topography of the Ahlergaarde catchment. The outlet station (82) and the upstream sub-catchment (21) are used in the study.

**Figure 3:** Climatology of the Ahlergaarde catchment. Values for precipitation (P), reference evapotranspiration ($E_{T0}$), streamflow (Q) and temperature (T) are monthly average values over the period 1990-2013.

**Figure 4:** Hydrographs for the 2000-2003 period. Percentage bias (PBias) and Nash-Sutcliffe efficiency score (NSE) are
computed using the daily observed-simulated values for the complete 1990-2013 period. The scatter plots represent the observed-simulated annual low flow values.

**Figure 5:** PBias and skill in terms of accuracy and sharpness of monthly means of daily streamflow of raw and preprocessed forecasts at station 82. The Y-axis represents the target month, and the X-axis represents the different lead times at which target months are forecasted. Values in blue range show a positive bias/skill and values in red a negative bias/skill. LS and
QM represent the bias for a streamflow forecast generated with bias adjusted forcing using the Linear Scaling-Delta Change and Quantile Mapping, respectively. Circles represent the cases where the distribution of the accuracy and/or sharpness for ESP differs from that of the ECMWF System 4 generated forecasts at a 5% significance level using the WMW-test.

**Figure 6:** Statistical consistency of monthly mean of daily streamflow forecasts for winter (upper row) and summer (bottom row) for station 82. LS and QM represent the consistency of a streamflow forecast generated with bias adjusted forcing using
the Linear Scaling and Quantile Mapping methods, respectively. Different colors represent different months in the season. The black lines parallel to the 1.1 diagonal are the Kolmogorov bands at the 5% significance level.





**Figure 7:** PBias and skill (sharpness and accuracy) of daily monthly mean streamflow forecasts for a post-processed forecast using the QM method for predictions generated using raw and preprocessed forcings from ECMWF System 4. Legend is the same as Figure 5.

**Figure 8:** Statistical consistency of daily monthly mean streamflow postprocessed forecasts for summer (upper row) and winter (bottom row) for station 82. The black lines parallel to the 1.1 diagonal are the Kolmogorov bands at the 5% significance level.

**Figure 9:** (a) and (b) Skill of accuracy (CRPSS) for upstream station 21 and outlet station 82 for target months Jan-Dec at lead time 4. Triangles and circles represent the forecasts generated with raw ECMWF System 4 forcings and preprocessed with LS, respectively, whereas black and blue lines represent the comparison against observed and simulated streamflow, respectively. The second row shows the monthly forecasts of December streamflow initialized in September (4 month lead time) for predictions using raw (c) and preprocessed (d) forcings for all years in the record (1990-2013) for station 21.

**Figure 10:** CRPSS of station 21 for forecasts generated with raw (a) and preprocessed (b) forcings, in addition to the postprocessed forecasts (c-d).

**Figure 11:** Low flow forecasts for the years 1990-2013 for generated using raw forcings (a), preprocessed forcings with the LS (c) and the QM (e) method and postprocessed streamflow for forecasts generated with raw (b) and preprocessed inputs (d and f). The forecasts are initiated in April. Blue shaded box-plots are ESP forecasts. CRPSS and SS are computed using Eq. (4) with ESP as reference.

**Figure A1:** Interpretation of the Probabilistic Integral Transform Diagram (PIT). Observed values are generated with a standard normal distribution $N(0,1)$. Instances of bias in both mean and dispersion are generated with the following distributions $N(1.5,1)$, $N(-1.5,1)$, $N(0,3)$ and $N(0,0.3)$ for an overestimated, underestimated, overdispersive and underdispersive system, respectively.





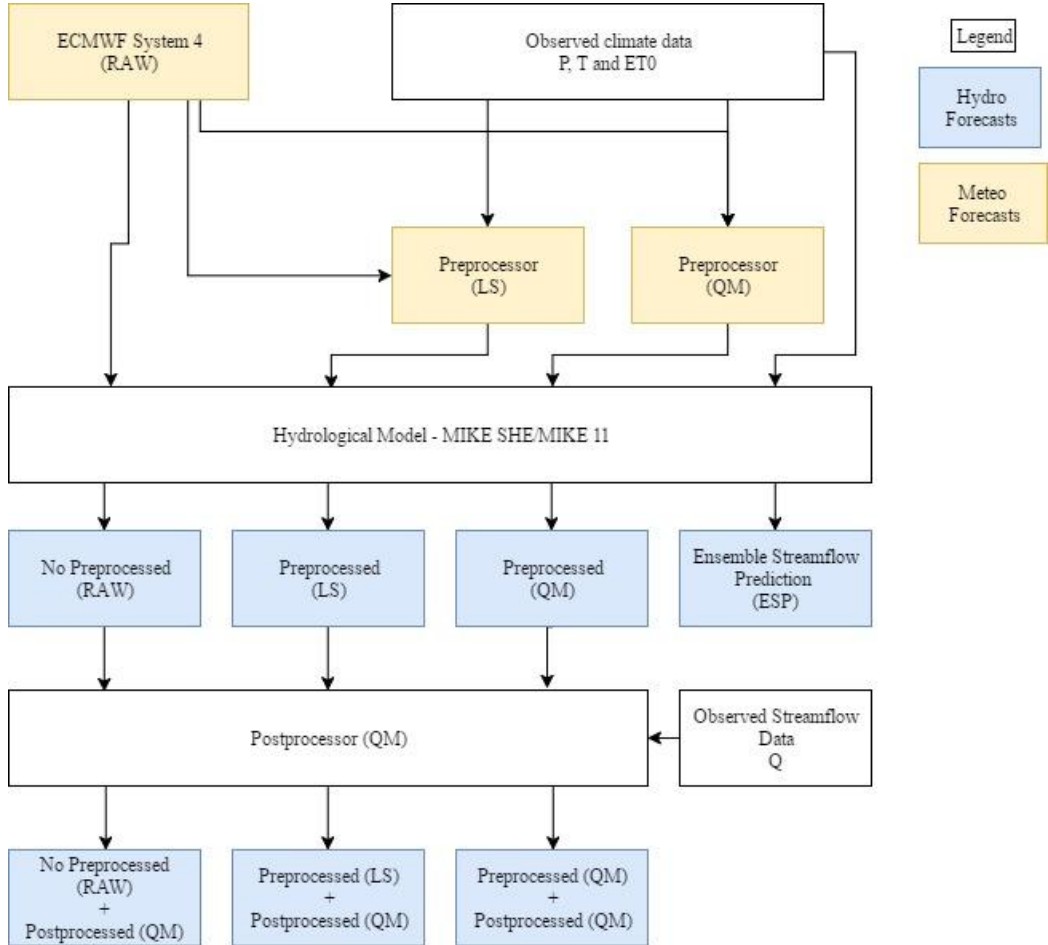

**Figure 1:** Diagram of generation of forecasts and verification procedures. RAW refers to the uncorrected ECMWF System 4

5    forecasts, while LS and QM refer to forecasts (either meteorological or hydrological) that are corrected using the Linear
Scaling/Delta Change or Quantile Mapping method, respectively, for precipitation (P), temperature (T) and reference
evapotranspiration ($E_{T0}$).



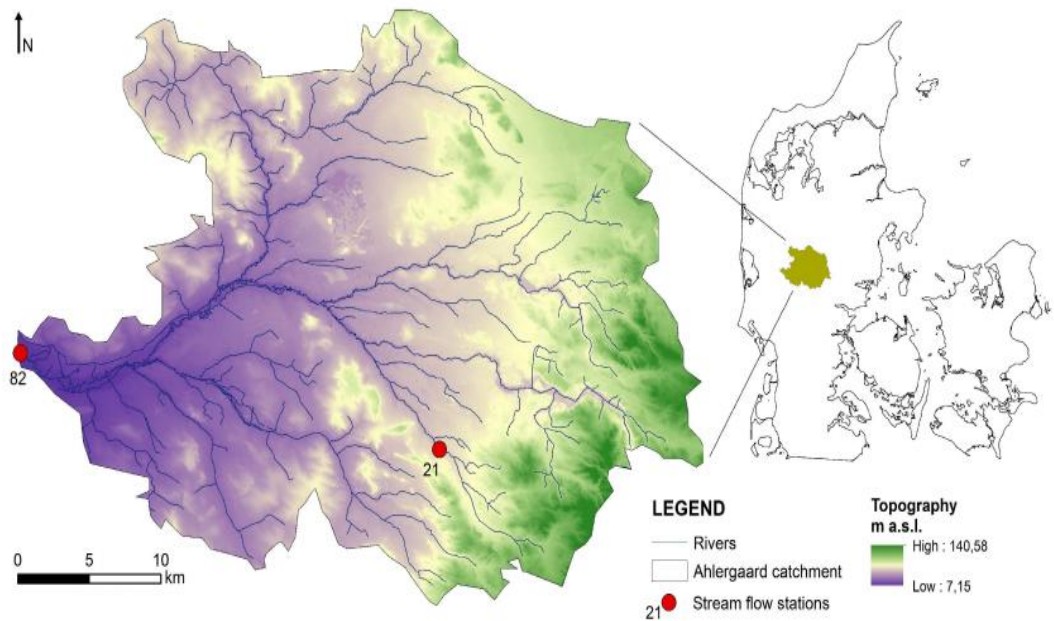

**Figure 2:** Location and topography of the Ahlergaarde catchment. The outlet station (82) and the upstream sub-catchment
(21) are used in the study.





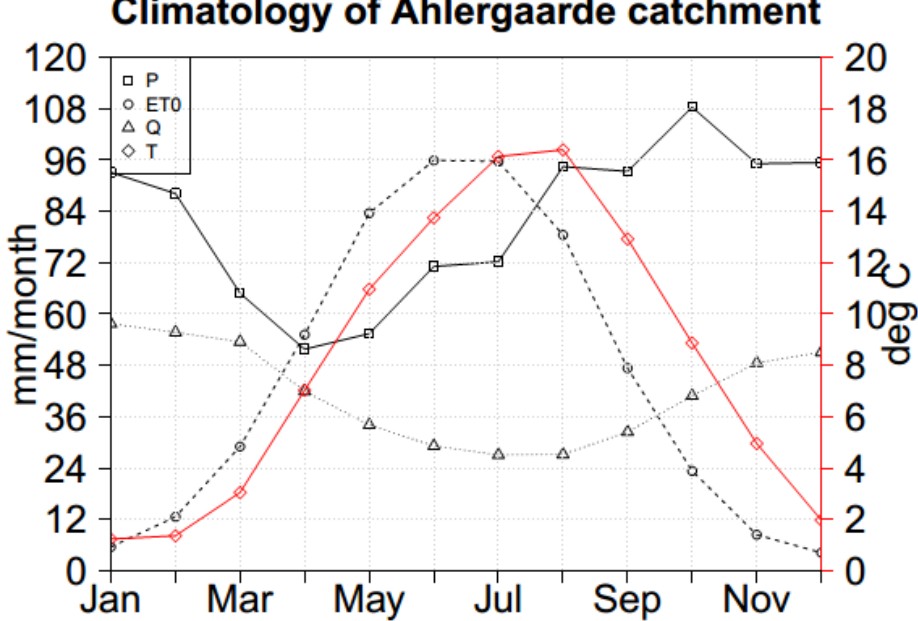

**Figure 3:** Climatology of the Ahlegaarde catchment. Values for precipitation (P), reference evapotranspiration ($E_{T0}$), streamflow (Q) and temperature (T) are monthly average values over the period 1990-2013.



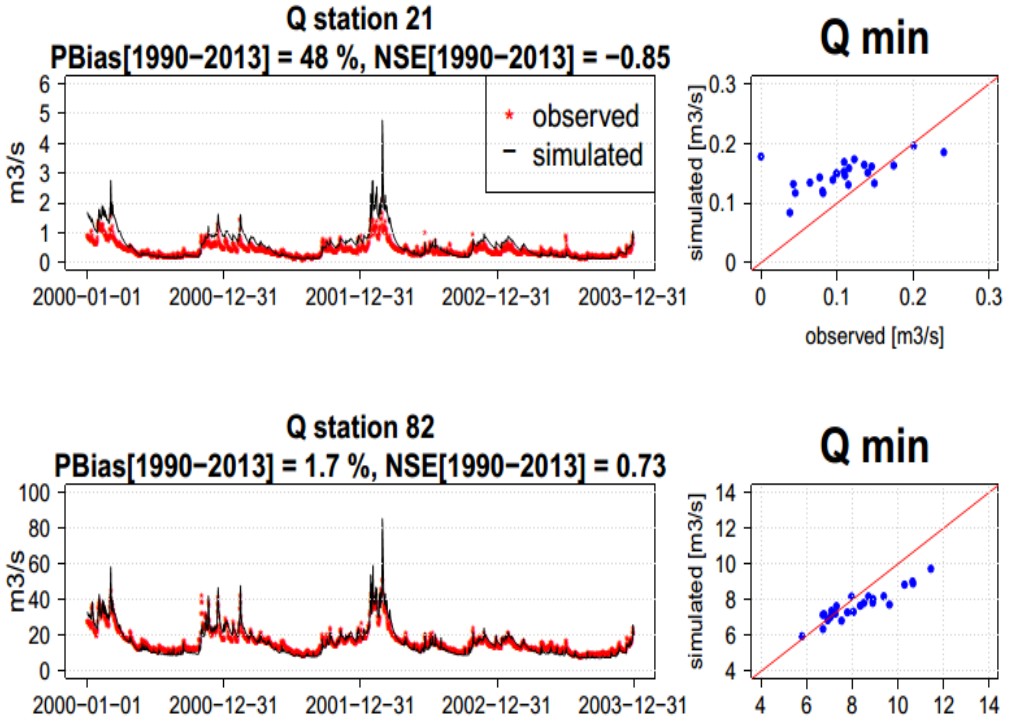

**Figure 4:** Hydrographs for the 2000-2003 period. Percentage bias (PBias) and Nash-Sutcliffe efficiency score (NSE) are computed using the daily observed-simulated values for the complete 1990-2013 period. The scatter plots represent the observed-simulated annual low flow values.




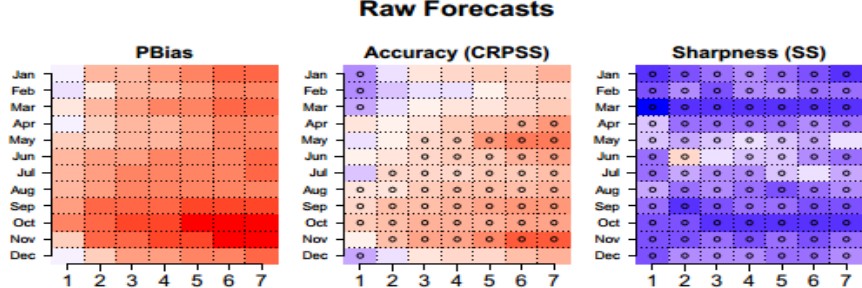

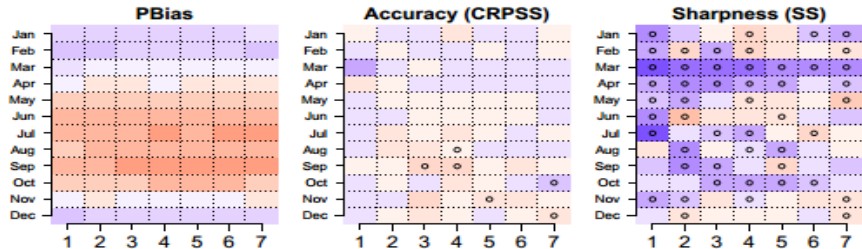

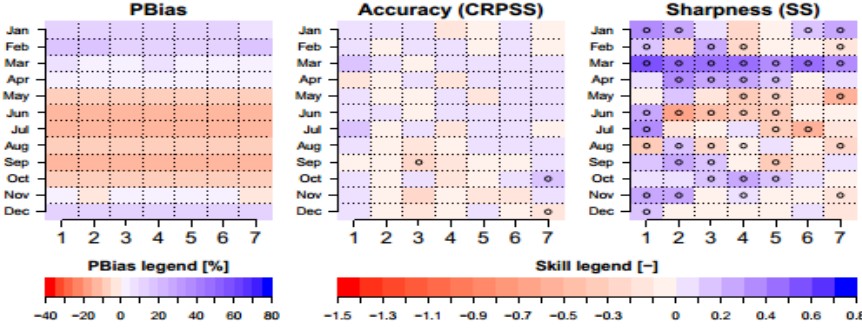

**Figure 5:** PBias and skill in terms of accuracy and sharpness of monthly means of daily streamflow of raw and preprocessed forecasts at station 82. The Y-axis represents the target month, and the X-axis represents the different lead times at which target months are forecasted. Values in blue range show a positive bias/skill and values in red a negative bias/skill. LS and QM represent the bias for a streamflow forecast generated with bias adjusted forcing using the Linear Scaling-Delta Change and Quantile Mapping, respectively. Circles represent the cases where the distribution of the accuracy and/or sharpness for ESP differs from that of the ECMWF System 4 generated forecasts at a 5% significance level using the WMW-test.





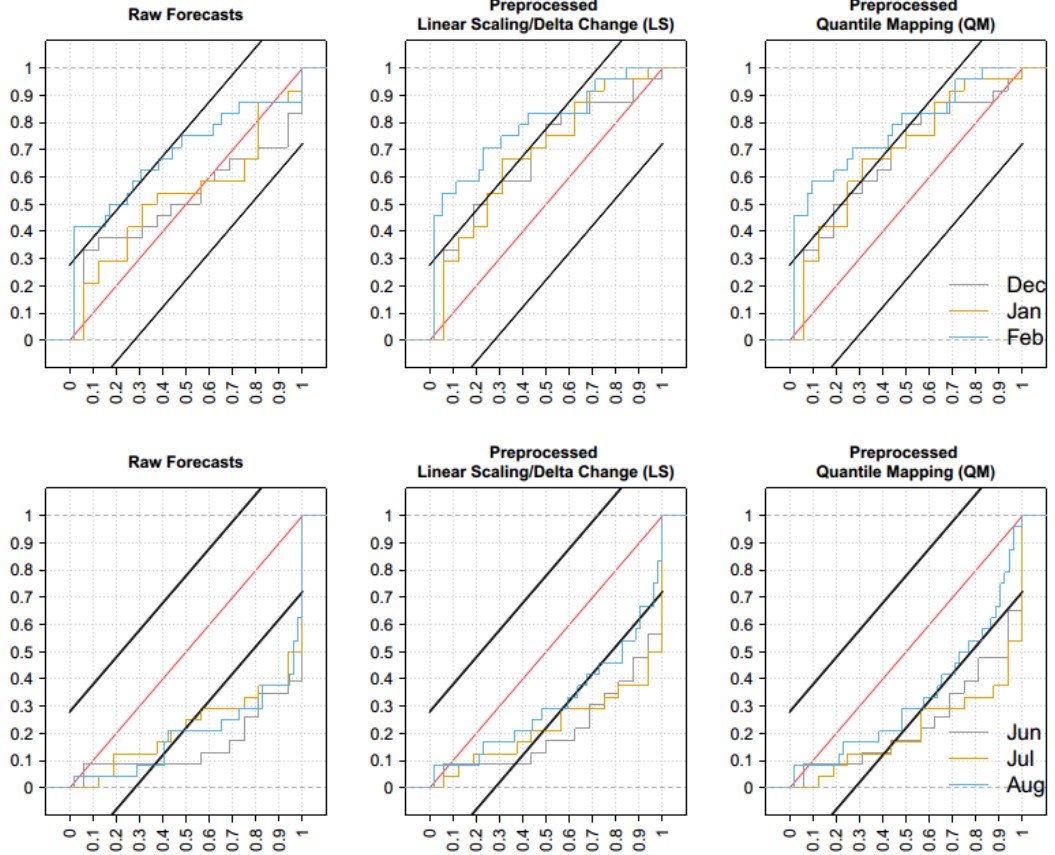

**Figure 6:** Statistical consistency of monthly mean of daily streamflow forecasts for winter (upper row) and summer (bottom row) for station 82. LS and QM represent the consistency of a streamflow forecast generated with bias adjusted forcing using the Linear Scaling and Quantile Mapping methods, respectively. Different colors represent different months in the season. The black lines parallel to the 1.1 diagonal are the Kolmogorov bands at the 5% significance level.




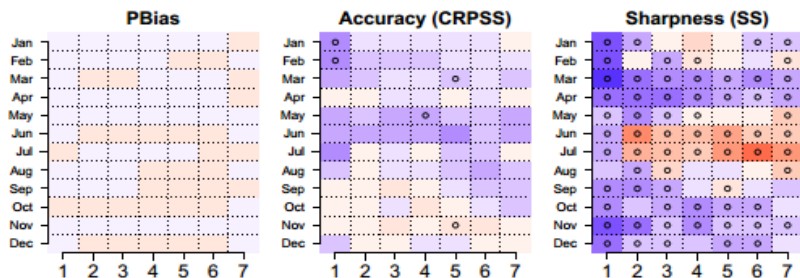

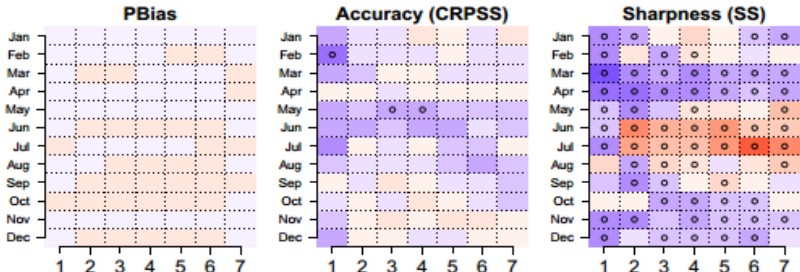

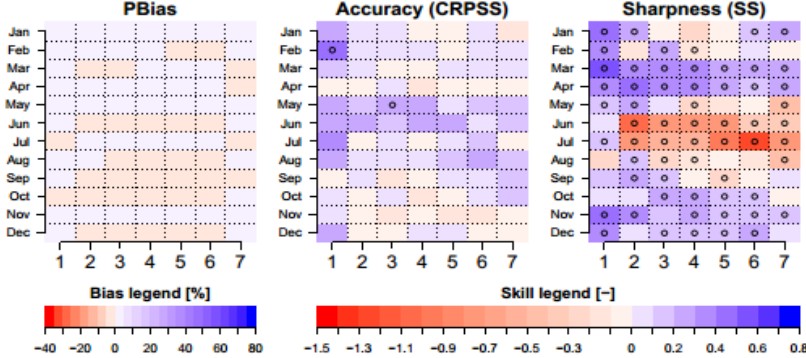

**Figure 7:** PBias and skill (sharpness and accuracy) of daily monthly mean streamflow forecasts for a post-processed forecast using the QM method for predictions generated using raw and preprocessed forcings from ECMWF System 4. Legend is the same as Figure 5.





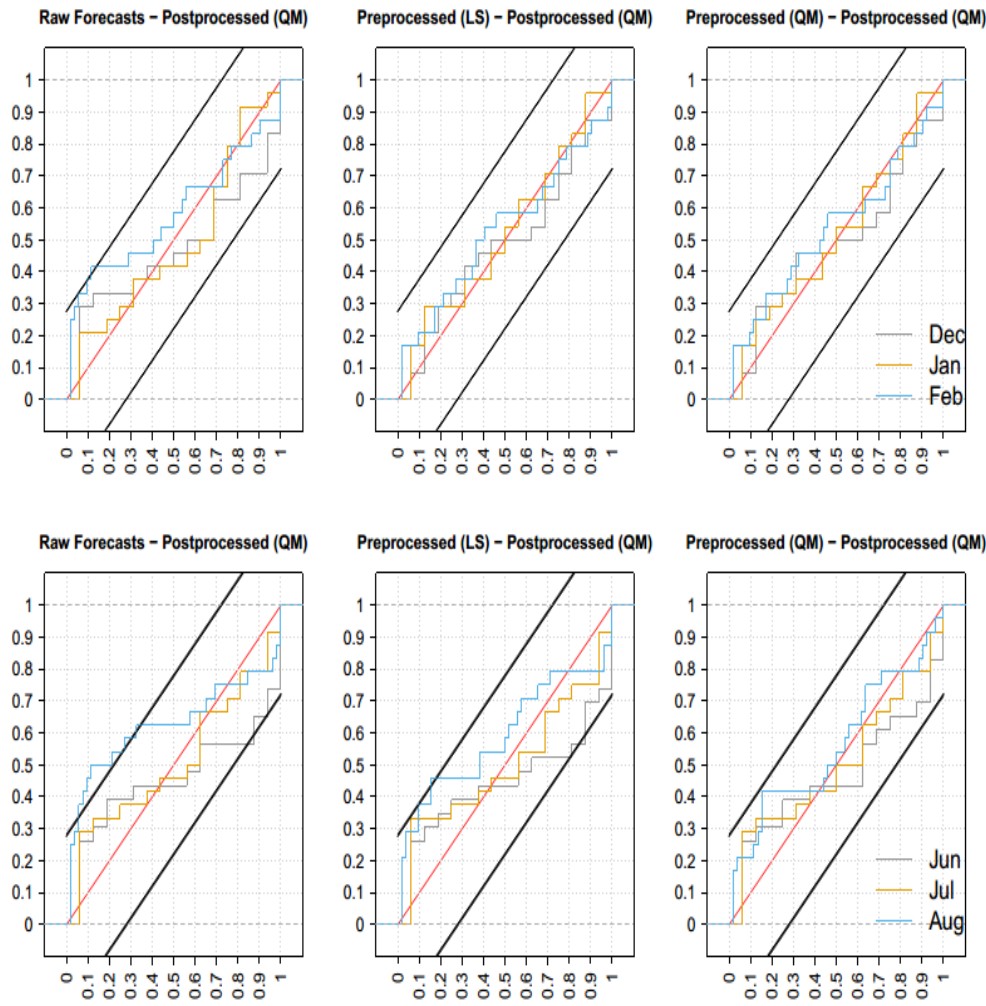

**Figure 8:** Statistical consistency of daily monthly mean streamflow postprocessed forecasts for summer (upper row) and winter (bottom row) for station 82. The black lines parallel to the 1.1 diagonal are the Kolmogorov bands at the 5% significance level.




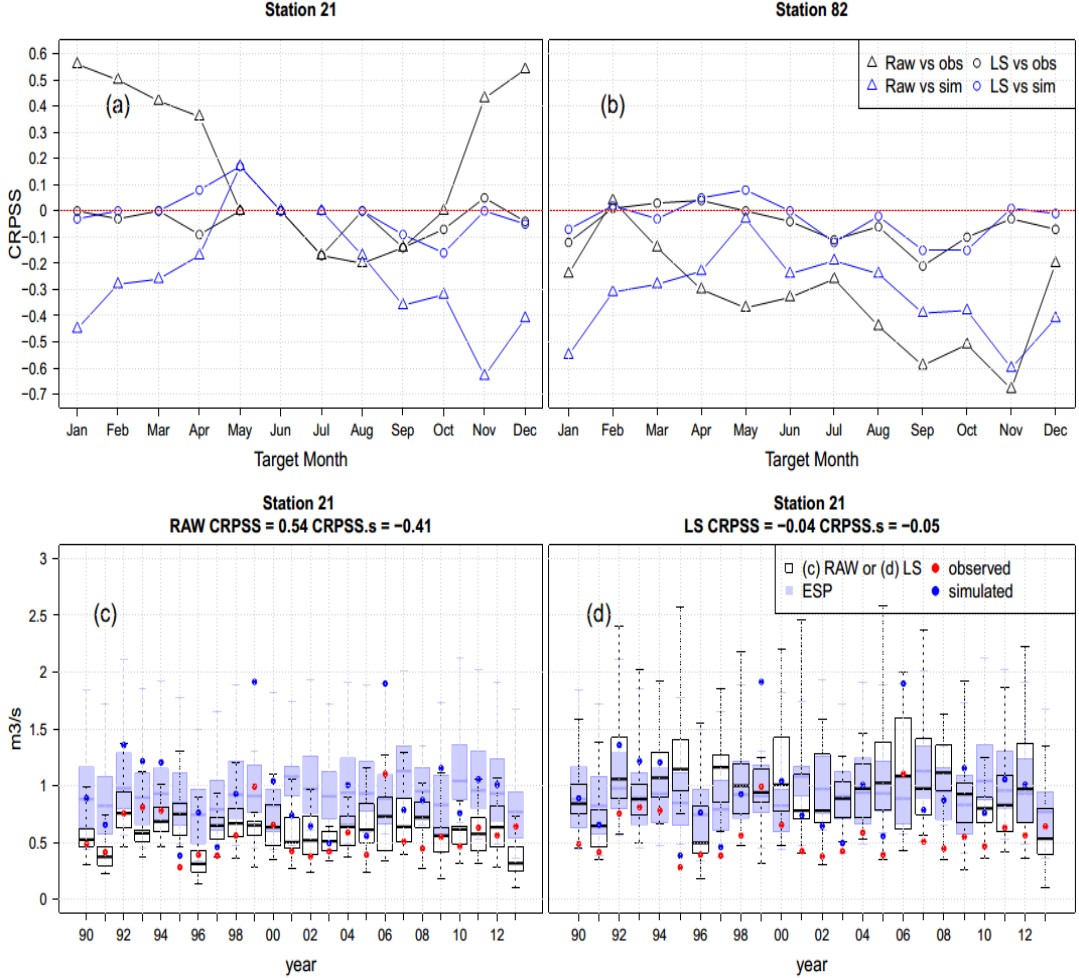

**Figure 9:** (a) and (b) Skill of accuracy (CRPSS) for upstream station 21 and outlet station 82 for target months Jan-Dec at lead time 4. Triangles and circles represent the forecasts generated with raw ECMWF System 4 forcings and preprocessed with LS, respectively, whereas black and blue lines represent the comparison against observed and simulated streamflow, respectively. The second row shows the monthly forecasts of December streamflow initialized in September (4 month lead time) for predictions using raw (c) and preprocessed (d) forcings for all years in the record (1990-2013) for station 21.





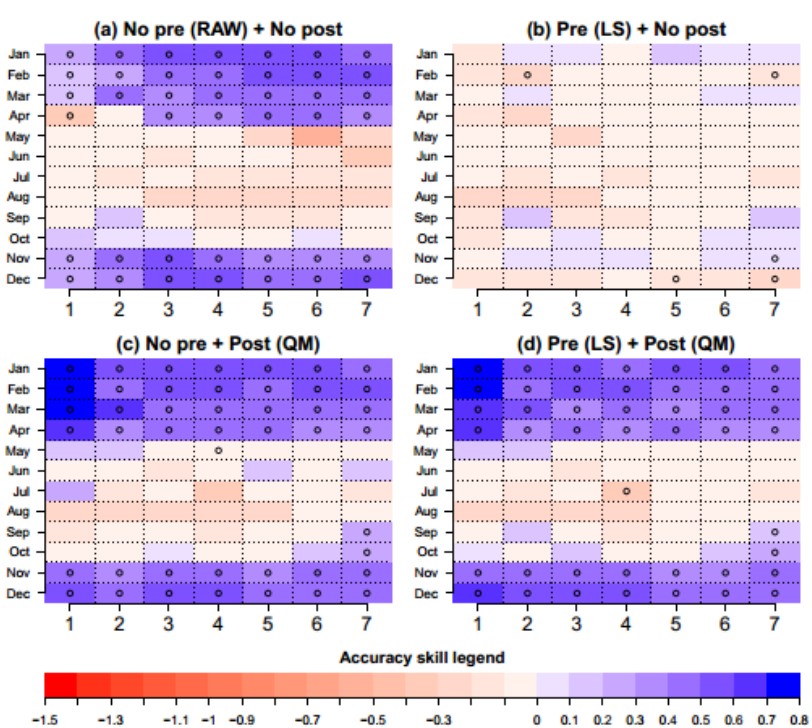

**Figure 10:** CRPSS of station 21 for forecasts generated with raw (a) and preprocessed (b) forcings, in addition to the postprocessed forecasts (c-d).





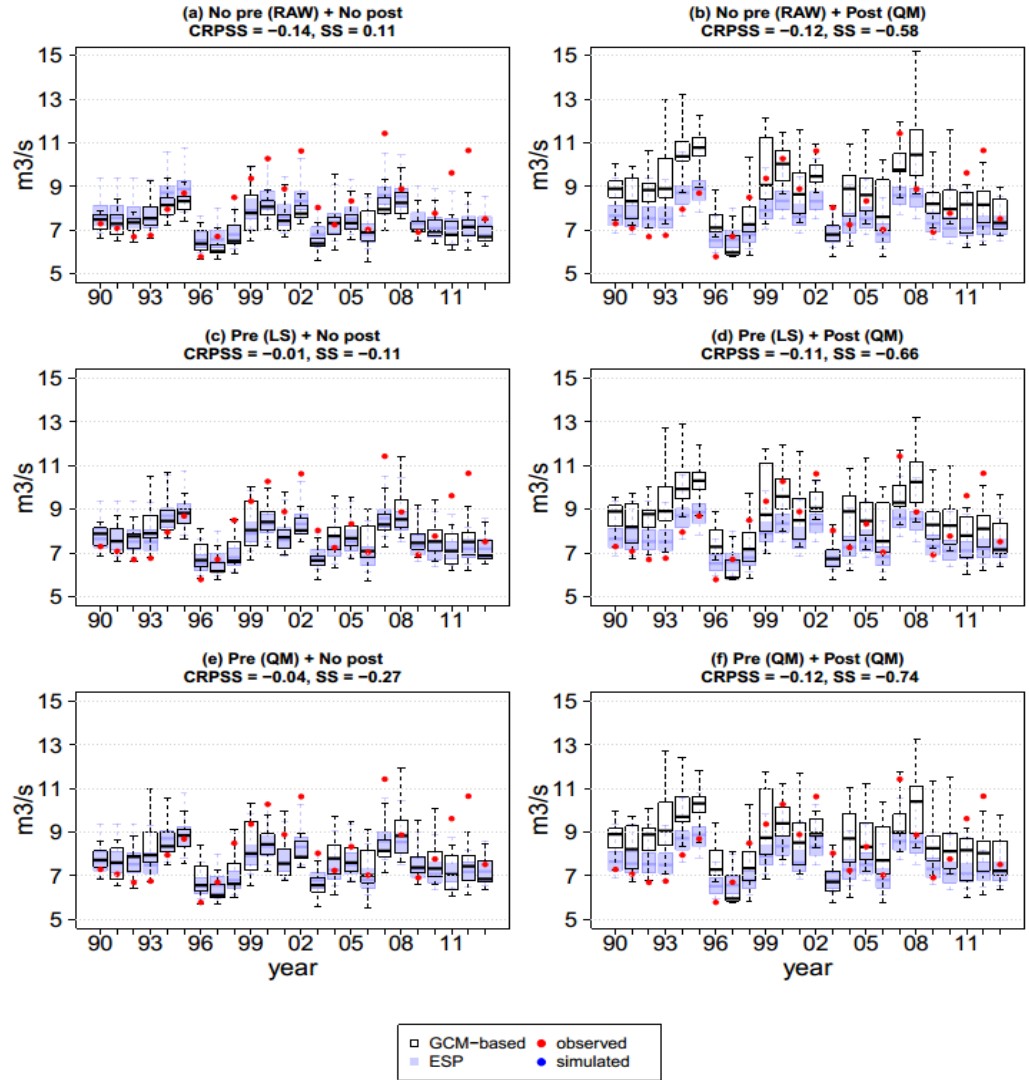

**Figure 11:** Low flow forecasts for the years 1990-2013 for generated using raw forcings (a), preprocessed forcings with the LS (c) and the QM (e) method and postprocessed streamflow for forecasts generated with raw (b) and preprocessed inputs (d and f). The forecasts are initiated in April. Blue shaded box-plots are ESP forecasts. CRPSS and SS are computed using Eq. (4) with ESP as reference.





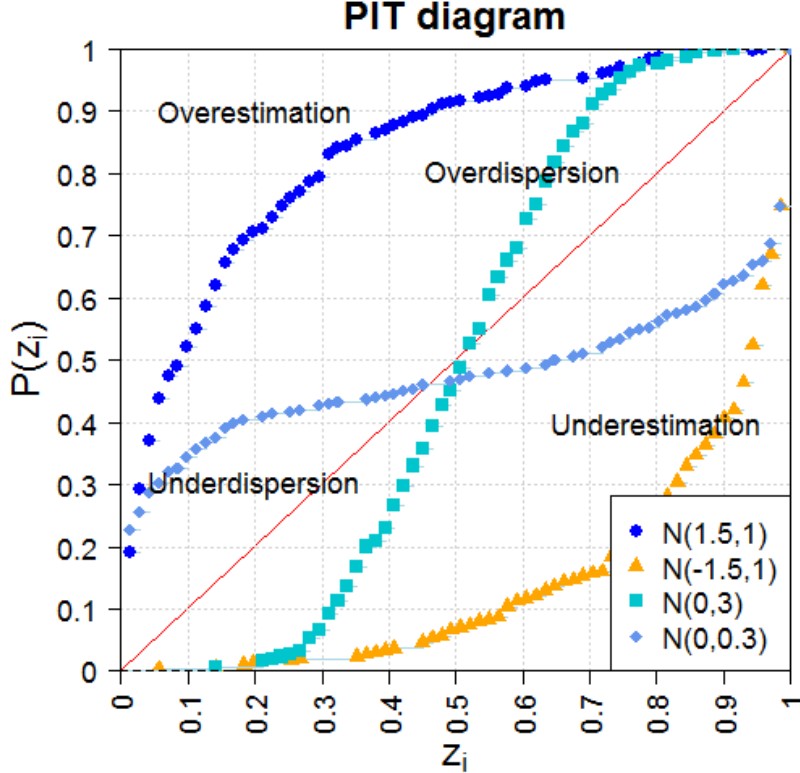

**Figure A1:** Interpretation of the Probabilistic Integral Transform Diagram (PIT). Observed values are generated with a standard normal distribution $N(0,1)$. Instances of bias in both mean and dispersion are generated with the following distributions $N(1.5,1)$, $N(-1.5,1)$, $N(0,3)$ and $N(0,0.3)$ for an overestimated, underestimated, overdispersive and underdispersive system, respectively.

