# Peer review of "Seasonal streamflow forecasts in the Ahlergaarde catchment, Denmark: effect of preprocessing and postprocessing on skill and statistical consistency"

_Hydrology and Earth System Sciences, 2017_

## Referee Comment (RC1) · Anonymous Referee #1 · 14 Aug 2017

General comments:

I found the paper interesting to read, and it addresses some relevant scientific questions within the field of hydrology and seasonal forecasting. The methodology is clearly outlined, and the overall presentation is well structured. However, I recommend that the introduction be expanded with more information about the use and skill of GCM-based seasonal forecasts in the region, where I found some information to be lacking. Also, the main scientific conclusion needs more clarification in my opinion. Finally, I suggest some minor revisions, for which I refer to the specific comments below.

[Figure]

The language is generally precise and understandable, albeit a bit verbose sometimes. I recommend maybe trying to shorten some sentences, and to use the active tense more instead of the passive tense. Finally, see also the technical corrections below to improve the English.

I do recommend the manuscript for publication, subject to minor revisions.

Specific comments

- p2, line 4: "perceived" : this gives the impression that the lack of reliability is a perception, do you mean a general observed fact here?

- p2, paragraph 4, and further on in the manuscript: you say that seasonal forecasting in the region is a "challenging endeavor". I would like some more information here on the state of the art and skill of current seasonal predictions for temperature, precipitation, .. in the region. I realize that results are/will be presented in a different paper (Lucatero et al), but since this is still "under preparation", it hinders me a bit in comprehending the general skill of the seasonal meteorological forecasts you are using as input for your hydrological model. Could you provide some salient features at least? It would also aid in understanding the conclusions better. If there is no skill past a certain forecast range for example, it is important to appreciate this before using the forecast as input... Is there a published reference that could be useful here?

- p.3, paragraph 3: concerning the DMI observations: I assume these are daily values? Please clarify. paragraph 4: does the hydrological model take snow melt into account? How important is snow in this area?

- p.3, section 2.2: you use acronyms here that haven't been explained yet (LS and QM). And as mentioned above, I would prefer to have some of the most important features of the (as yet unavailable) Lucatero et al paper available here.

- p.4, line 1: why the change in the number of ensemble members?

- p.4, section 2.3: could you add an existing publication referencing to the Quantile

Mapping method?

- p.5, concerning PIT diagrams. Is there a reason that you use these instead of e.g. verification rank (Talagrand) histograms? Since you are comparing ensemble members with observations (and not PDF's with observations), it seems less appropriate... Also, please clarify how you go from the ensemble to the CDF. Is it just the empirical CDF? Or do you use some smoothing, like fitting a gaussian to each ensemble (cfr. Grimit and Mass, 2004).

- p.6, section 2.5. Low flow forecasting. You evaluate these in the same way as the monthly flow forecasts. Could you comment on timing errors? For example, how relevant would it be for water management if the forecast predicts the correct low flow, but in the wrong month? You also mention two other studies on low flows that exist. Is there a link with their results and yours? Do they use the same verification methodology?

- p.9, Summary and Conclusions. I am missing a bit a general conclusion that could be useful for end users of the hydrological forecasts. Are the current seasonal meteorological forecasts just not good enough? Could better postprocessing techniques ameliorate the situation? Data assimilation? The last paragraph seems very important, concerning catchment characteristics and taking advantage of hydrological memory.. but seems almost added as an afterthought. A few final sentences on the "best way forward" according to the authors could help here, and if a more explicit link to the results on seasonal forecasts of Lucatero et al could be made, this could provide a clearer overview in my opinion.

Technical Corrections

- p.1 line 11: forecasts (plural) line 37: "or outputs from, use..." : not grammatically correct, please rephrase

- p.2 line 7: "should" instead of "may" ? line 25: "of ECMWF" instead of "from" line 31: "How do... compare to those..." (plural)

[Figure]

- p.4 line 2: should be "number of ensemble members"

- p.5 line 13: "Moreover..." -> this sentence needs to be rephrased line 15: should be "raw OR preprocessed" ?

- p.9 line 28: "effect on" instead of "in" line 33: "Thus, it seems..." instead of "It thus"

- p.10 line 13: "did not perform" instead of "did not performed" line 16: 'small' and 'large' errors (instead of 'high' and 'low') line 23: should be "..which in turn helps.." line 29: "particularly for" line 30: "which in turn translate"" please rephrase

---

## Referee Comment (RC2) · Anonymous Referee #1 · 14 Aug 2017

Two additional comments:

- Figure 8 caption should explicitly mention "PIT diagrams" - The reference "Molteni et al, 2011" does not include the journal (ECMWF Technical Memorandum 656)

[Figure]

---

## Referee Comment (RC3) · M. Zappa (Referee) · 29 Aug 2017

[referee-annotated manuscript omitted]

---

## Author Comment (AC1) · 12 Oct 2017

We highly appreciate the insightful comments that the reviewers have provided. Their comments and their suggestions to modifications will surely improve the quality of the paper.

**Reviewer # 1**

**General comments:**

**I found the paper interesting to read, and it addresses some relevant scientific questions within the field of hydrology and seasonal forecasting. The methodology is clearly outlined, and the overall presentation is well structured.**

**Comment 1. However, I recommend that the introduction be expanded with more information about the use and skill of GCM based seasonal forecasts in the region, where I found some information to be lacking. Also, the main scientific conclusion needs more clarification in my opinion. Finally, I suggest some minor revisions, for which I refer to the specific comments below.**

Thank you for bringing this issue up. We realize that connections to the companionship paper "Lucatero, D., Madsen, H., Refsgaard, J. C., Kidmose, J., and Jensen, K. H.: On the skill of raw and postprocessed ensemble seasonal meteorological forecasts in Denmark, Hydrol. Earth Syst. Sci. Discuss., https://doi.org/10.5194/hess-2017-366, in review, 2017" are lacking. We will make the connection more clearly in the updated manuscript version.

**Comment 2. The language is generally precise and understandable, albeit a bit verbose sometimes. I recommend maybe trying to shorten some sentences, and to use the active tense more instead of the passive tense. Finally, see also the technical corrections below to improve the English.**

We will improve the writing to ease the reading.

**I do recommend the manuscript for publication, subject to minor revisions.**

**Specific comments**

**Comment 3. - p2, line 4: "perceived" : this gives the impression that the lack of reliability is a perception, do you mean a general observed fact here?**

It is an observed fact (Weisheimer and Palmer, 2014). We will change the sentence accordingly to avoid confusion.

**Comment 4. -p2, paragraph4, and further on in the manuscript: you say that seasonal forecasting in the region is a "challenging endeavor". I would like some more information here on the state of the art and skill of current seasonal predictions for temperature, precipitation, .. in the region. I realize that results are/will be presented in a different paper (Lucatero et al), but since this is still "under preparation", it hinders me a bit in comprehending the general skill of the seasonal meteorological forecasts you are using as input for your hydrological model. Could you provide some salient features at least? It would also aid in understanding the conclusions better. If there is no skill past a certain forecast range for example, it is important to appreciate this before using the forecast as input... Is there a published reference that could be useful here?**

We will clarify the connection to the companion paper as stated before.

**Comment 5. -p.3, paragraph3: concerning the DMI observations: I assume these are daily values? Please clarify. paragraph 4: does the hydrological model take snow melt into account? How important is snow in this area?**

Yes, DMI observations are daily values, which will be clarified in the revised version of the manuscript. The model takes snowmelt into account by using a simple degree-day model formulation. Overall, snow processes are not important in the study area.

**Comment 6. -p.3,section2.2: you use acronyms here that haven't been explained yet (LS and QM). And as mentioned above, I would prefer to have some of the most important features of the (as yet unavailable) Lucatero et al paper available here.**

The comments are noted and they will be incorporated in the revised version.

**Comment 7. - p.4, line 1: why the change in the number of ensemble members?**

This is the data we received from the meteorological forecast provider (ECMWF). We assume that the increase of ensemble size for February, May, August and November is done with the objective of increasing quality of the forecasts of the upcoming season, for example summer (JJA) for forecasts initialized in May. We will clarify if this is the case in the updated version of the manuscript.

**Comment 8. - p.4, section 2.3: could you add an existing publication referencing to the Quantile Mapping method?**

We will add a reference on Quantile Mapping (Zhao et al., 2017).

**Comment 9. - p.5, concerning PIT diagrams. Is there a reason that you use these instead of e.g. verification rank (Talagrand) histograms? Since you are comparing ensemble members with observations (and not PDF's with observations), it seems less appropriate... Also, please clarify how you go from the ensemble to the CDF. Is it just the empirical CDF? Or do you use some smoothing, like fitting a gaussian to each ensemble (cfr. Grimit and Mass, 2004).**

First, we make use of PIT diagrams for purely practical graphical reasons. We considered that it would be easier for the reader to visualize lines in Fig. 6 and Fig. 8 rather than bar plots in rank histograms. Moreover, we presume that if rank histograms were used instead, the conclusions would not change due to the connection between the rank histogram and the PIT diagram.  In the updated version of the paper, we will make an attempt to demonstrate this or discuss the implications and appropriateness of our choice (PIT diagrams over rank histograms ).

Related to the above and answering to your second question, we go from the ensemble to the CDF using empirical CDF. $z_i = P(X<=y_i)$ is the number of ensembles below or equal to the observed value divided by the number of ensembles (Page 5, lines 23-25). We will also clarify this in the new version of the manuscript.

**Comment 10. - p.6, section 2.5. Low flow forecasting. You evaluate these in the same way as the monthly flow forecasts. Could you comment on timing errors? For example, how relevant would it be for water management if the forecast predicts the correct low flow, but in the wrong month? You also mention two other studies on low flows that exist. Is there a link with their results and yours? Do they use the same verification methodology?**

Observed low flow is computed as the flow of the day with the minimum discharge over the seven month forecasting period (April-October). Forecasted low flow (for each ensemble) is computed in the same manner.

Timing errors will only be visible if forecasted low flow was chosen to be the discharge values of the day where low flow was observed, which is not the case here.

We will make the connection to the cited literature more clearly in the updated version of the manuscript.

**Comment 11. - p.9, Summary and Conclusions. I am missing a bit a general conclusion that could be useful for end users of the hydrological forecasts. Are the current seasonal meteorological forecasts just not good enough? Could better postprocessing techniques ameliorate the situation? Data assimilation? The last paragraph seems very important, concerning catchment characteristics and taking advantage of hydrological memory.. but seems almost added as an afterthought. A few final sentences on the "best way forward" according to the authors could help here, and if a more explicit link to the results on seasonal forecasts of Lucatero et al could be made, this could provide a clearer overview in my opinion.**

These comments are highly valid and we will improve the conclusions and discussion accordingly.

**Technical Corrections**

**- p.1 line 11: forecasts (plural) line 37: "or outputs from, use..." : not grammatically correct, please rephrase - p.2 line 7: "should" instead of "may" ? line 25: "of ECMWF" instead of "from" line 31: "How do... compare to those..." (plural)**

**- p.4 line 2: should be "number of ensemble members"**

**- p.5 line 13: "Moreover..." -> this sentence needs to be rephrased line 15: should be "raw OR preprocessed" ?**

**- p.9 line 28: "effect on" instead of "in" line 33: "Thus, it seems..." instead of "It thus"**

**- p.10 line 13: "did not perform" instead of "did not performed" line 16: 'small' and 'large' errors (instead of 'high' and 'low') line 23: should be "..which in turn helps.." line 29: "particularly for" line 30: "which in turn translate"" please rephrase**

**Two additional comments:**

**- Figure 8 caption should explicitly mention "PIT diagrams"**

**- The reference "Molteni et al, 2011" does not include the journal (ECMWF Technical Memorandum 656)**

All technical corrections will be incorporated in the revised manuscript.

Weisheimer, A. and Palmer, T. N.: On the reliability of seasonal climate forecasts, J. R. Soc. Interface, 11, 20131162, 2014.

Zhao, T., Bennett, J., Wang, Q. J., Schepen, A., Wood, A., Robertson, D. and Ramos, M.-H.: How suitable is quantile mapping for post-processing GCM precipitation forecasts?, J. Clim., JCLI-D-16-0652.1, doi:10.1175/JCLI-D-16-0652.1, 2017.

**Reviewer # 2**

**Dear authors,**

**This paper is a nice well designed study on pre- and post-processed seasonal ensemble predictions. Such studies are needed to learn the limits and opportunities yielded by such prediction systems.**

**Comment 1. In some parts the referencing and discussion suffers from the fact that a companion paper dealing with the meteorological pre-processing is also in revision. The authors should better state that there is a companion paper and therefore some aspects concerning meteorology are not dealt here.**

This issue was also brought up by Reviewer # 1. In the updated version of the manuscript we will improve the connection to the companion paper.

**Comment 2. While it is nice to have a study dealing with both pre- and post-processing. The applied methodologies (mainly for the post-processing) are rather simple as compared to state-of-the-art methods. The authors should give more room on the discussion of their findings with respect to findings obtained by "higher-order" post-processing techniques.**

We are aware that perhaps more sophisticated processing methods might have an impact on the quality of the forecasts. However, this comparison should be done in a consistent manner, which is a research question that is outside of the scope of this study. Our objective was to evaluate the benefits of pre- and post-processing using a fairly simple and popular method. We will address more clearly this issue on the Discussion section of the updated manuscript.

**Comment 3. The issue of having in some cases 15 and in other cases 51 members should be addressed also in the results and discussion sections. I give in the commented manuscript a reference to have a look at.**

We will carry out the evaluation as suggested . It will be addressed in the new version.

**Comment 4. The discussion section should be extended and separated from the conclusions.**

We will separate the discussion and conclusion sections in the revised manuscript and generally improve these two sections.

We respond to specific questions noted in the pdf of the manuscript in the following.

**Comment 5 .Page 1. Line 21. Where do you see it in your outcomes?**

We do not observe it here. However, we believe this to be an important component of the objective of the study (improving forecast quality) which can be dealt with in future research.

**Comment 6 .Page 3. Line 3. How representative is this basin within Denmark and the Baltic region?**

It represents a groundwater dominated catchment dominated by outwash coarse sandy materials representative of the western part of Denmark. Further it is located in a temperate climate and as such represents the Baltic region.

**Comment 7. Page 3. How is estimated ET0 by DMI? How is estimated ET0 in the ECMWF products? How does this relate to the DMI product?**

Both DMI observations and forecasts are estimated using the Makkink equation as explained in Page 3 Lines 39-40.

**Comment 8. Page 4. Any reason for that? Wouldn't be easier to have 15 all the time? Can you put in all figure an asterisk on the months where you have 51 members?**

As explained in the response of Comment 7 of Reviewer #1, the data we received from the reforecast provider has different ensemble members for the individual months. We will emphasize this in the revised manuscript and indicate in the figures when the analysis is based on 51 members.

**Comment 9. Page 4. Do I understand right, all experiment start on January 1990 and run until the start of the forecast eg: 1.10.1993 or 1.4.2002 ... does the system not allow to save states?**

The model is run once from January 1990 up until December 2014 saving initial conditions on the first day of each month.

**Comment 10. Page 6. In the presented data for the year 2000 most issues on gauge 21 seems to be related to the cold season. Can you comment/confirm? The large bias could be also a sign of some sub-surface transfer of water in the headwater regions?**

Yes, the bias is higher for the cold season. If there is some sub-surface transfer of water this should be accounted for in the model.

**Comment 11. Page 6. The quality of raw forecasts (CRPSS) gradually decrease with lead time. This is easy to catch and agrees with our findings concerning monthly forecasts (See Fundel et al. paper cited before). Now, if I look at the CRPSS obtained after applying the pre-processors I cannot find any intuitive pattern in the skill of your system concerning lead time, while a pattern related to the season of initialization is intelligible using the bias metric. So there is good news here that the metrics improve, but at first glance one can think that this is not linked to processes. Can you comment?**

We need to investigate this and will make comments on it in the revised manuscript.

**Comment 12. Page 7 and 8. Can you please include in your reply and exemplary forecast with hydrograph and members of the different version you use? Also here it would be nice to see in the reply an example of post-processed forecast with the raw ensemble as a background.**

[Figure]

**Figure 1:** Example of a forecast of daily streamflow (initialized 01-10-2000, outlet station 82) generated with different strategies. Pre and Post stand for preprocessing and postprocessing, respectively. RAW, LS, QM and ESP stand for raw forecast, linear scaling , quantile mapping and Ensemble Streamflow Prediction respectively.

**Comment 13. Page 7. Would be of course nice to have also a figure for one month, and different lead times, same applies to Figure 8**

[Figure]

**Figure 2:** PIT diagrams of target month December at different lead times (LT) for different forecast strategies. For example, LT 3 corresponds to the December forecast initialized in October.

**Comment 14. Page 8. Page 12. This is the companion paper in HESSD, isn't?**

Yes, this is the one. As stated above, we will make the connection clearer on the revised version of the manuscript.

**Comment 15. Page 16. As a curiosity, have you tried to Postprocess ESP?**

No, this was not the purpose of the study.

**Comment 16. Page 20. Can be that February works bests, because you have 51 members in the raw forecasts? How would this look like if you pick up 15 random members several times (bootstrapping) and then you average it?**

We believe it has more to do with compensational errors during winter than the difference in number of ensemble members as discussed in Sect. 3.5. However, in order to remove the effect of varying ensemble size, we will make the analysis as you suggest and add the resulting changes (if there are any) in the revised version of the manuscript.